# QueST: Querying Functional and Structural Niches on Spatial Transcriptomics Data via Contrastive Subgraph Embedding

## Abstract

The functional or structural spatial regions within tissues, referred to as spatial niches, are elements for illustrating the spatial contexts of multicellular organisms. A key challenge is querying shared niches across diverse tissues, which is crucial for achieving a comprehensive understanding of the organization and phenotypes of cell populations. However, current data analysis methods predominantly focus on creating spatial-aware embeddings for cells, neglecting the development of niche-level representations for effective querying. To address this gap, we introduce QueST, a novel niche representation learning model designed for querying spatial niches across multiple samples. QueST utilizes a novel subgraph contrastive learning approach to explicitly capture niche-level characteristics and incorporates adversarial training to mitigate batch effects. We evaluate QueST on established benchmarks using human and mouse datasets, demonstrating its superiority over state-of-the-art graph representation learning methods in accurate niche queries. Overall, QueST offers a specialized model for spatial niche queries, paving the way for deeper insights into the patterns and mechanisms of cell spatial organization across tissues.

## 1 Introduction

A spatial niche in tissue sections is a cell and its surroundings with specific functionality or structure. Cells within tissues operate in complex spatial niches, where their functions are significantly influenced by interactions with neighboring cells. These niches are characterized by the spatial arrangement of cells and the molecular signals exchanged among them, playing a crucial role in regulating higher-order tissue functions such as immune responses, tissue regeneration, and disease progression (Wu et al., 2021; Meylan et al., 2022; Palla et al., 2022). Importantly, the behavior of the same cell type can vary dramatically depending on its niche (Wagner et al., 2016), underscoring the necessity of understanding the spatial context in cellular behavior.

Recent advances in spatial transcriptomics technologies (Xia et al., 2019; Eng et al., 2019; Rao et al., 2020; Rodriques et al., 2019; Chen et al., 2022) have enabled scientists to inspect this important biological component from a computational view, as these technologies manage to generate gene expression data for each cell or sequencing spot while preserving its location information. Some researchers have endeavored to develop context-aware representation learning methods on spatial transcriptomics data (Hu et al., 2021; Dong & Zhang, 2022; Long et al., 2023). However, these methods focus on cell or node embedding clustering on a single spatial transcriptomics sample, limiting their application to studying spatial niches. Also, most methods are unable to remove the batch effect, limiting their usability on multiple samples.

Extending the analysis of spatial niches across multiple samples amplifies our ability to discern universal patterns and unique variations that single-sample studies might overlook. By integrating spatial transcriptomics data across different samples, including, different tissues, developmental stages, or disease conditions, we may uncover how similar niches or microenvironmental features manifest in diverse biological contexts, shedding light on the underlying mechanisms that drive normal physiology and pathological states. Recent researchers have made a step towards this multiple sample integration target (Zeira et al., 2022; Xia et al., 2023; Zhou et al., 2023), which are the

so-called 'alignment' methods. However, these methods focus on cross-sample similarity on the cell-cell level rather than considering it on a niche level. Although the cell-cell alignment results may generally indicate niche similarity structure, they cannot directly serve as niche similarity measurement. Furthermore, these methods are limited to aligning only two samples per training run, making them impractical for analyzing large cohorts of spatial transcriptomics samples.

Querying spatial niches across different samples presents several unique challenges. First, explicit representations are needed to compute the similarities among spatial niches and perform query tasks. However, existing graph deep learning methods typically focus on node embedding rather than subgraph embedding. Some methods (Wu et al., 2022; Fischer et al., 2022; Wang et al., 2023) have tried to represent a subgraph via posing a pooling layer or readout function on the node embedding, but this simple approach cannot guarantee proper utilization of information within and between subgraphs under the self-supervised setting. Second, the gene expression data between different samples may contain considerable batch effect (Korsunsky et al., 2019), which must be removed before comparing niches on a sample-wise level.

To address the above challenges, we present QueST, a novel niche representation learning model that explicitly learns biologically meaningful representations for spatial niches and can perform spatial niche queries across multiple samples. We construct benchmarks for the niche query problem with the Human Dorsolateral Prefrontal Cortex (DLPFC) dataset (Maynard et al., 2021) and the Mouse Olfactory Bulb Tissue (MOBT) dataset (Guo et al., 2023), and evaluate QueST against several state-of-the-art graph representation learning and alignment methods on these benchmarks. Our framework makes the following key contributions:

- **Novel Niche Representation Learning Strategy:** We introduce a novel graph contrastive learning strategy to explicitly learn the representation of a niche, which includes obtaining subgraph embedding via a pooling layer and partially shuffling the graph while leaving certain parts fixed to generate positive and negative niche pairs.

- **Adversarial Batch Effect Removal:** We borrow the idea of Adversarial Auto-encoder (Makhzani et al., 2015) and introduce an adversarial training strategy to remove batch information from the model's latent space during the training stage and empower the encoder with the ability to remove batch effect from unseen samples.

## 2 TASK FORMULATION

Consider a spatial transcriptomics dataset $\mathcal{D}$ that includes gene expression measurements for $N$ cells, $G$ genes, and $T$ samples. For each cell $i$, we have access to the gene expression vector $x_i \in \mathbb{R}^G$, its spatial location $s_i \in \mathbb{R}^2$, and a one-hot batch annotation $b_i \in \mathbb{N}^T$ indicating the sample from which cell $i$ originates.

We construct a spatial graph $G = (V, E)$ based on the spatial proximity relationship among cells within each sample, where each node has node features that correspond to a cell's gene expression vector. We define the $k$-hop subgraph centered at cell $i$ as its corresponding spatial niche, denoted as Niche($i$). Consequently, the dataset $\mathcal{D}$ will contain up to $N$ niches in total. We categorize these niches into two sets:

**Query niche set** $\mathcal{Q}$: a small set of niches that are well studied, possessing known properties of interest, such as specific cell type patterns.

**Reference niche set** $\mathcal{R}$: a larger set of niches where the properties of interest remain largely unknown and are worth exploration.

Now we introduce the main objective of this paper:

**Niche Query Problem: What is the most similarity niche in $\mathcal{R}$ to a query niche in $\mathcal{Q}$?**

Figure 1 provides a schematic illustration of this problem. Each query niche corresponds to a "similar niche(s)" area on a reference sample, where a series of niches will all serve as the correct answers to the **Niche Query Problem**. We aim for the model not only to identify a single correct niche within the "similar niche(s)" area but also to reveal this entire area.

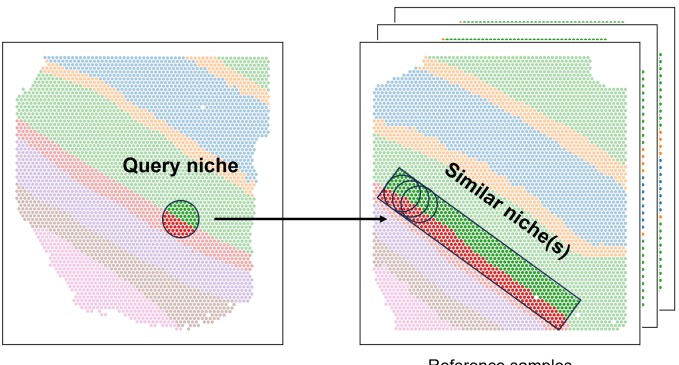

Figure 1: Illustration of the Niche Query Problem

Assume $\text{Niche}(q) \in \mathcal{Q}$ is a given query niche and $\text{Niche}(r) \in \mathcal{R}$ is an arbitrary niche from the reference set. The objective of the niche query problem is:

$$\text{Niche}(\hat{r}) = \underset{\text{Niche}(r) \in \mathcal{R}}{\arg\max} \; \text{sim}_{\mathcal{M}} \left[ \text{Niche}(q), \text{Niche}(r) \right] \quad (1)$$

where $\text{sim}_{\mathcal{M}}(\cdot, \cdot)$ is a pairwise similarity function defined by the model $\mathcal{M}$.

We then define several metrics to evaluate $\mathcal{M}$'s performance from the following two aspects:

**Best Niche Match Accuracy:** This perspective examines whether $\text{Niche}(\hat{r})$ is accurate, requiring a ground truth metric for pairwise niche similarity. Since no existing niche similarity measurement is available, we derive this metric from ground truth cell type annotations, transforming the niche similarity problem into a subgraph similarity problem. We adopt the Wasserstein Weisfeiler-Lehman (WWL) Graph Kernel method (Togninalli et al., 2019) to calculate the similarity value between two arbitrary subgraphs. This method involves multiple information aggregation iterations, where each vertex aggregates the node embeddings (initialized with cell type labels) from its neighbors and hashes the aggregated label into a new embedding for itself. This process can be defined as:

$$\boldsymbol{a}^{h+1}(v) = \text{hash}\left(\boldsymbol{a}^h(v), \mathcal{N}^h(v)\right) \quad (2)$$

After these iterations, we compute the Wasserstein distance between the node embedding matrices of the two niches, reversing it to establish the ground truth similarity measure:

$$\text{sim}_{\text{wwl}} \left[ \text{Niche}(q), \text{Niche}(\hat{r}) \right] = 1 - \text{WassersteinDistance} \left[ \text{Niche}(q), \text{Niche}(\hat{r}) \right] \quad (3)$$

**Overall Similarity Accuracy:** This perspective assesses the accuracy of $\text{sim}_{\mathcal{M}}$, which in essence reflects the model's ability to reveal the entire "similar niche(s)" region in Figure 1. We compare the similarity results between $\text{sim}_{\text{wwl}}$ and $\text{sim}_{\mathcal{M}}$ across the entire sample. Given a query niche $\text{Niche}(q)$ and a reference sample $\mathcal{T}$ with its corresponding niche set $\text{Niche}(\mathcal{T}) = \{\text{Niche}(r) | r \in \mathcal{T}\}$, each similarity function $\text{sim}$ generates a similarity vector $S$, with each entry representing the similarity value:

$$S_{\text{sim}}[r] = \text{sim} \left[ \text{Niche}(q), \text{Niche}(r) \right] \quad (4)$$

We then evaluate the agreements between $S_{\text{sim}_{\text{wwl}}}$ and $S_{\text{sim}_{\mathcal{M}}}$ using the Pearson correlation metric:

$$\text{Accuracy}_{\text{sim}_{\mathcal{M}}} = \text{PearsonCorrelation} \left( S_{\text{sim}_{\text{wwl}}}, S_{\text{sim}_{\mathcal{M}}} \right) \quad (5)$$

## 3 THE QUEST APPROACH

In this section, we present QueST, the self-supervised graph auto-encoder framework, to query niches on multi-sample spatial transcriptomics data based on graph neural networks. Besides the main graph auto-encoder architecture, QueST has a contrastive learning module and an adversarial training module, which are two critical components that distinguish QueST from previous approaches. The contrastive learning module enables the model to discriminate between niches with different cell type compositions and topological structures, while the adversarial training module removes the batch effect from the latent niche representation. We present a schematic illustration of QueST model architecture in Figure 2.

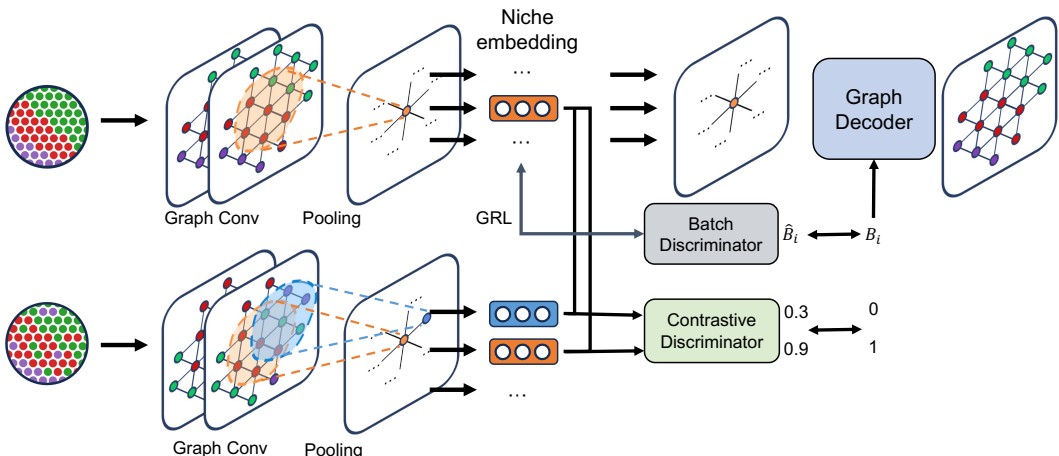

Figure 2: QueST model architecture

## 3.1 CONTRASTIVE NICHE REPRESENTATION LEARNING

The main architecture of QueST uses a graph auto-encoder with GIN (Xu et al., 2018). The graph encoder maps the normalized gene expression vector to a $d$ dimensional latent space:

$$\boldsymbol{h}_i^{(l+1)} = \text{MLP}^{(l)}\left((1+\epsilon) \cdot \boldsymbol{h}_i^{(l)} + \sum_{j \in \mathcal{N}(i)} \boldsymbol{h}_j^{(l)}\right) \tag{6}$$

where $\boldsymbol{h}_i^{(l)}$ is the node embedding of cell $i$ after the $l$-th encoder layer of GNN, with $\boldsymbol{h}_i^{(0)}$ initialized as $\boldsymbol{x}_i$. $l_{\text{enc}}$ is the number of layers of the encoder, and $\boldsymbol{z}_i = \boldsymbol{h}_i^{(l_{\text{enc}})} \in \mathbb{R}^d$ is the node embedding of cell $i$ after the encoder. $\epsilon$ is a scalar that controls the weight of each node's own feature in the aggregation process.

However, this typical node-embedding fashion only learns a representation for each node or cell rather than for each niche. Even if the node embedding considers the k-hop neighborhood via information aggregation in Graph Neural Networks, the signal faints as the distance from the central node increases, while each cell should be treated equally within a niche when the prior knowledge of importance for each cell is absent. To emphasize this equal information aggregation within a niche, we add a pooling layer to force the model to learn the unbiased representation for the entire subgraph or niche centered at each cell:

$$\boldsymbol{z}_{\text{niche}(i)} = \text{POOL}\left(\boldsymbol{z}_j | j \in \mathcal{N}(i)\right) \tag{7}$$

where $\boldsymbol{z}_{\text{niche}(i)}$ is the embedding of the niche centered at cell $i$. We then introduce a typical graph decoder $f_{\text{dec}}$ and a reconstruction loss to make sure that the embedding contains enough information to reconstruct the input:

$$L_{\text{recon}} = \frac{1}{N} \sum_{i=1}^{N} \left\| f_{dec}\left(\boldsymbol{z}_{\text{niche}(i))}\right) - \boldsymbol{x}_i \right\|^2 \tag{8}$$

To make sure the niche embedding encodes both the local and global topological and structural information, we introduce a contrastive learning framework to enhance the quality of the niche embedding. Consider the spatial graph $G_k = (V_k, E_k)$ for sample $k, k = 1, ..., T$. During each training epoch, before $G_k$ goes through the model, we conduct data augmentation via the following steps:

**1. Niche Fixing:** we randomly sample a certain subset of nodes $V_{\text{fix}} \subset V$. For each niche with a central node in $V_{\text{fix}}$, we generate a positive and a negative pair in the next step. Denote this fixed niche set as $Niche_{\text{fix}} = \{\text{Niche}(v) | v \in V_{\text{fix}}\}$.

**2. Graph Corruption:** We generate a corrupted graph by randomly shuffling node features while preserving the graph structure. During this process, the nodes included by $Niche_{\text{fix}}$ will have their node features fixed, while the rest of the nodes will randomly exchange their node feature vectors. Denote the corrupted graph as $G'_k = (V'_k, E'_k)$. For each node $v \in V$, denote $v'$ as its corresponding node in the corrupted graph $G'$.

**3. Dual Encoding:** Once the corrupted graph $G'$ is generated, both $G$ and $G'$ will be fed into the model and will have its own latent niche representation denoted as $z_{\text{Niche(v)}}$ and $z_{\text{Niche(v')}}$.

**4. Contrastive Sampling:** After obtaining $z_{\text{Niche(v)}}$ and $z_{\text{Niche(v')}}$, for each node $v$ in $V_{\text{fix}}$ and its corresponding niche, we can define following the positive and negative embedding pair:

- Positive pair: $(z_{\text{Niche(v)}}, z_{\text{Niche(v')}})$;

- Negative pair: $(z_{\text{Niche(v)}}, z_{\text{Niche(u')}})$, where $u'$ is randomly sampled from $V' \backslash \{v\}$.

After the data augmentation step, we define the following contrastive loss:

$$L_{\text{contrast}} = \frac{1}{|V_{\text{fix}}|} \sum_{v \in V_{\text{fix}}} \log \left[ f_{\text{contrast}} \left( z_{\text{Niche(v)}}, z_{\text{Niche(v')}} \right) \right] + \log \left[ 1 - f_{\text{contrast}} \left( z_{\text{Niche(v)}}, z_{\text{Niche(u')}} \right) \right] \quad (9)$$

where we build the contrastive loss following Deep Graph Infomax (Velickovic et al., 2019), with $f_{\text{contrast}}$ being a bilinear discriminator $f_{\text{contrast}} : \mathbb{R}^d \times \mathbb{R}^d \to \mathbb{R}$ which distinguishes the positive pair from the negative pair.

## 3.2 BATCH REMOVAL VIA ADVERSARIAL TRAINING

The above architecture is designed to learn niche representation on one single spatial transcriptomics sample. To enable QueST query niches across multiple samples, we develop an adversarial training method to remove the batch effect from the latent niche representation when training the model on multiple samples. The basic logic of this adversarial training is to make a discriminator to discriminate batch labels from the niche embeddings while simultaneously training the encoder adversarially to fool the batch discriminator. To achieve this, we input the latent niche representation $z_{\text{Niche(v)}}$ to a batch discriminator $f_{\text{batch}}$ to predict which sample this niche is from and use a Gradient Reversal Layer (Ganin et al., 2016) for giving reverse gradient to the encoder. We denote the predicted batch labels as

$$\hat{b}_v = f_{\text{batch}} \left( GRL \left( z_{\text{Niche(v)}} \right) \right) \quad (10)$$

where GRL is:

$$GRL \left( z_{\text{Niche(v)}} \right) = z_{\text{Niche(v)}}; \quad \frac{\partial GRL \left( z_{\text{Niche(v)}} \right)}{\partial z_{\text{Niche(v)}}} = -1 \quad (11)$$

then we have the following batch prediction loss:

$$L_{\text{batch}} = -\frac{1}{N} \sum_{i=1}^{N} \left( \sum_{j=1}^{T} b_i^{(j)} \log \hat{b}_i^{(j)} + \left( 1 - b_i^{(j)} \right) \log \left( 1 - \hat{b}_i^{(j)} \right) \right) \quad (12)$$

However, the removal of batch information from the latent representation will add difficulties to reconstruction, since the gene expression that which decoder is required to reconstruct still contains the batch effect. To mitigate the negative impact of batch removal on the reconstruction task, we add batch information back to the input of the decoder:

$$x_{\text{recon}_v} = f_{\text{dec}}(z_{\text{Niche(v)}}, b_v) \quad (13)$$

and the reconstruction loss becomes

$$L_{\text{recon}} = \frac{1}{N} \sum_{i=1}^{N} \| x_{\text{recon}_i} - x_i \|^2 \quad (14)$$

### 3.3 QueST Overall Loss Function

The overall loss function of QueST is defined as follows:

$$L = \lambda_1 L_{\text{recon}} + \lambda_2 L_{\text{contrast}} + \lambda_3 L_{\text{batch}} \tag{15}$$

where we set $\lambda_1 = 1, \lambda_2 = 1, \lambda_3 = 0.1$ as default across all experiments. For other hyperparameters and implementation details, please refer to section A.3.

### 3.4 Querying Niches in Latent Space

Finally, we measure the similarity between two arbitrary spatial niches using the cosine similarity of their latent representation, which is used for the query:

$$\text{sim}_{\text{QueST}}[\text{Niche}(q), \text{Niche}(r)] = \frac{\boldsymbol{z}_{\text{Niche(q)}}^{\top} \boldsymbol{z}_{\text{Niche}(r)}}{\left\| \boldsymbol{z}_{\text{Niche(q)}} \right\| \cdot \left\| \boldsymbol{z}_{\text{Niche}(r)} \right\|} \tag{16}$$

## 4 Experiments

In this section, we present our experimental results. We set $k = 3$ as the default for the definition of a spatial niche across all experiments. For the WWL Graph Kernel, We set the iteration number to be 3 and use it as the ground truth similarity measurement.

### 4.1 Benchmark Construction

As there are currently no available multi-sample datasets with niche-level annotations that can serve as benchmarks, we synthesize our niche query benchmarks based on existing spatial transcriptomics datasets with cell-type annotation.

Specifically, we use two widely used datasets in spatial transcriptomics: the Human Dorsolateral Prefrontal Cortex (DLPFC) dataset and the Mouse Olfactory Bulb Tissue (MOBT) dataset. These datasets represent niche query problems with varying levels of challenges. On each dataset, we first separate samples into query and reference set, and then generate niches that need to be queried. The DLPFC dataset consists of 12 samples from the human brain, with ground truth cell type annotation illustrated in Figure A.1. We partition this dataset into a query set with 1 sample and a reference set with 11 samples. This setting requires the model to perform batch effect removal and niche query on a large cohort of spatial transcriptomics samples. The MOBT dataset contains 3 samples from different sequencing technologies including 10X Visium, Stereo-seq (Chen et al., 2022) and Slide-seq V2 (Stickels et al., 2021). The radius of data spots for each technology differs; they are $50\mu m, 35\mu m$ and $10\mu m$, respectively. As the ground truth cell type annotation shown in Figure A.2, the variation in spot resolution affects the spatial distribution of the sequencing spots or cells, which is an even harder task. It requires the niche query methods not only to remove the batch effect in gene expression but also to model the spatial context on heterogeneous graphs. We split the MOBT dataset into a query set with 1 sample from Stereo-seq and a reference set with 2 samples from 10X Visium and Slide-seq V2. The detailed description of these two datasets can be found in section A.1.

Once the query and reference sample sets are established, any niche within the query sample can correspond to a niche query problem. In this paper, we generate several query niches by changing these two parameters:

**1. Size:** This refers to the number of cells included in the niche. We consider three scenarios with the query niche containing 50, 100, and 200 cells.

**2. Cell Type Pattern:** This refers to the number of cell types contained in the niche. We also examine three scenarios, where the query niche includes 1, 2, and 3 cell types.

By combining these two aspects of variation, in total 9 query niches can be defined for each query sample, as illustrated in Figure A.3 and Figure A.4. See section A.2 for detailed generation of query niches.

We evaluate QueST's performance against three popular graph deep learning methods in spatial transcriptomics: GraphST, SLAT, and STAGATE. GraphST and STAGATE are representation learning

Table 1: Results for Pearson correlation on DLPFC dataset. We report the mean value on the 11 reference samples for each query niche.

| Niche Name | GraphST | SLAT | STAGATE | QueST |
|---|---|---|---|---|
| Layer3_Layer4_Layer5_100 | 0.476 | **0.528** | 0.130 | 0.383 |
| Layer3_Layer4_Layer5_200 | 0.477 | 0.607 | 0.102 | **0.620** |
| Layer3_Layer4_Layer5_50 | 0.335 | 0.506 | 0.068 | **0.585** |
| Layer4_100 | 0.288 | 0.418 | 0.062 | **0.535** |
| Layer4_200 | 0.285 | 0.430 | 0.063 | **0.535** |
| Layer4_50 | 0.282 | 0.415 | 0.058 | **0.529** |
| Layer5_Layer6_100 | 0.386 | 0.658 | 0.132 | **0.802** |
| Layer5_Layer6_200 | 0.463 | 0.729 | 0.152 | **0.743** |
| Layer5_Layer6_50 | 0.433 | **0.720** | 0.140 | 0.714 |

methods designed for single samples, utilizing GCN (Kipf & Welling, 2016) and GAT (Veličković et al., 2017) as their respective model backbones, while SLAT is tailored for the alignment of two spatial transcriptomics slides, employing LGCN (Xia et al., 2023) as its backbone and SVD pre-processing for batch effect removal. Additionally, SLAT solves an optimal transport optimization problem in its latent space to establish node correspondence relationships between samples as alignment results. However, all these competing methods focus on node embedding learning; thus, we manually perform pooling on their embeddings to obtain niche representations. We utilize the cosine similarity between the representations of niches in reference samples and the query niche as the similarity measurement corresponding to these representation learning methods.

### 4.2 QUEST QUERIES NICHES ACCURATELY ON LARGE COHORTS OF REFERENCE SAMPLES

We first evaluate QueST's performance on the DLPFC datasets. Table 1 presents the performance of GraphST, STAGATE, SLAT, and QueST on the DLPFC dataset, measured by the Overall Similarity Accuracy. As shown, the niche similarity results generated by QueST closely resemble the ground truth, achieving the highest Pearson correlation for 7 out of the 9 query niches. SLAT, utilizing its LGCN and batch correction module, also performs well, attaining the best results for the remaining 2 query niches. In contrast, the representation learning methods designed for single samples, including GraphST and STAGATE, perform poorly on this niche query task.

Figure 3 and Figure A.5-A.9 showcase the spatial distribution of cosine similarity for the example query niche Layer5_Layer6_100. As shown in these figures, QueST visually aligns most closely with the ground truth, with the high-similarity regions closely following the ground truth patterns, exhibiting fewer noisy artifacts and a smoother gradient across the reference sample. This indicates that QueST excels at reproducing the similarity structure of the ground truth. In contrast, the other methods capture the general spatial similarity pattern but lack niche embedding learning, making their latent spaces unsuitable for accurately measuring niche similarity.

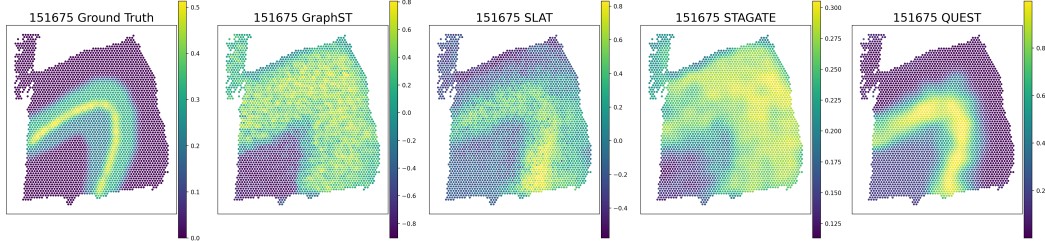

Figure 3: Visualization of cosine similarity of niche Layer5_Layer6_100 on sample 151675

Table 2 presents the performance of these methods based on **Best Niche Match Accuracy**. From this table, it is evident that QueST achieves the best results for nearly all query niches. These results are consistent with those from the **Overall Similarity Accuracy**, as a more reasonable similarity measurement is more likely to accurately address the **Niche Query Problem**. Figure 4 visualizes

Table 2: Results for Subgraph Wasserstein Distance on DLPFC dataset. We report the mean value on the 11 reference samples.

| Niche Name | GraphST | SLAT | STAGATE | QueST |
|---|---|---|---|---|
| Layer3_Layer4_Layer5_100 | 0.917 | **0.809** | 0.936 | 0.870 |
| Layer3_Layer4_Layer5_200 | 0.840 | 0.842 | 0.896 | **0.837** |
| Layer3_Layer4_Layer5_50 | 0.941 | 0.869 | 0.957 | **0.858** |
| Layer4_100 | 0.946 | 0.907 | 0.989 | **0.883** |
| Layer4_200 | 0.967 | 0.910 | 0.989 | **0.897** |
| Layer4_50 | 0.938 | 0.891 | 0.989 | **0.889** |
| Layer5_Layer6_100 | 0.758 | 0.803 | 0.969 | **0.751** |
| Layer5_Layer6_200 | 0.797 | 0.776 | 0.963 | **0.771** |
| Layer5_Layer6_50 | 0.781 | 0.782 | 0.968 | **0.775** |

the niche query results for sample 151675, highlighting only the queried region's cell types. We observe that QueST generates results that most closely match the results generated by ground truth similarity measurement, accurately capturing both the correct cell types and their adjacency relationships within the marked region. We also observe that alignment methods based on OT, such as SLAT, can find regions with cell type largely correct, but fail to produce a concentrated query area that corresponds to a local neighborhood or niche. This limitation arises because OT-based methods focus on computing cell-cell similarity rather than niche-niche similarity, mapping each cell in the query niche to its most similar counterpart in the reference niche. Therefore, while popular in multi-sample integration, these methods are not well-suited for the niche query task.

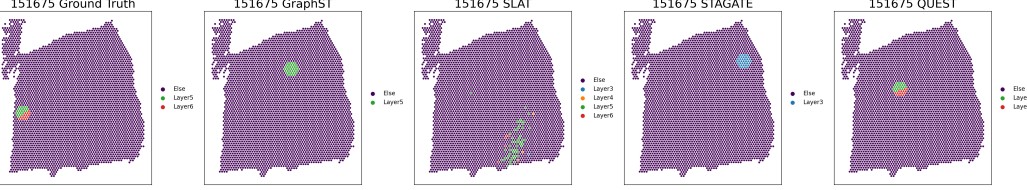

Figure 4: Visualization of query result on example niche Layer5_Layer6_100. We show the cell type of the region which is determined by the method as the query result and mask the cell type of the rest part.

### 4.3 QUEST QUERIES NICHE ACCURATELY ACROSS DIFFERENT SEQUENCING TECHNOLOGIES

Next, we evaluate QueST's performance on the Mouse Olfactory Bulb Tissue dataset. Table 3 shows the performance on the MOBT dataset in terms of the **Overall Similarity Accuracy**. Again, QueST achieves the highest Pearson correlation value on most of the query niches, demonstrating QueST's superiority for querying niches across samples with varying sequencing technologies. Figure 5 and Figure A.10-A.14 showcase the spatial distribution of cosine similarity of an example query niche GL_ONL_100. We observe that most of the results degrade significantly compared to those on DLPFC datasets, largely due to the substantial variations between samples introduced by the difference in sequencing technology. Nevertheless, QueST successfully retains most of the underlying similarity structure. In contrast, other methods suffer badly from this technical challenge, producing cosine similarity that almost spreads uniformly across the entire tissue, with low contrast between the target regions and the others.

Similar trends are observed when assessing these methods on the MOBT dataset using **Best Niche Match Accuracy**. Table 4 summarizes the mean performance of the nine query niches across two reference samples, with QueST consistently delivering the best results for most query niches. Figure 6 visualizes the query results for the example niche GL_ONL_100, again showing that QueST's outputs most closely resemble the ground truth.

Table 3: Results for Pearson Correlation on Mouse Olfactory Bulb Tissue dataset. We report the mean value on the 2 reference samples.

| Niche Name | GraphST | SLAT | STAGATE | QueST |
|---|---|---|---|---|
| GCL_100 | 0.437 | 0.769 | 0.576 | **0.777** |
| GCL_150 | 0.518 | 0.669 | 0.539 | **0.771** |
| GCL_50 | 0.359 | 0.747 | 0.551 | **0.773** |
| GCL_MCL_EPL_100 | 0.130 | 0.515 | 0.215 | **0.544** |
| GCL_MCL_EPL_150 | 0.116 | **0.572** | 0.283 | 0.557 |
| GCL_MCL_EPL_50 | 0.223 | **0.541** | 0.081 | 0.288 |
| GL_ONL_100 | 0.211 | 0.573 | -0.256 | **0.766** |
| GL_ONL_150 | 0.156 | 0.691 | -0.237 | **0.768** |
| GL_ONL_50 | 0.241 | 0.663 | -0.428 | **0.812** |



Figure 5: Visualization of cosine similarity on example niche GL_ONL_100

## 4.4 QUEST ABLATION EXPERIMENTS

In this section, we present the results of the ablation experiments, as shown in Table 5. From these results, we can draw several important observations:

First, removing the contrastive learning module significantly degrades the model's performance. This finding is intuitive, as the primary purpose of incorporating contrastive learning is to help the model distinguish between potentially similar yet fundamentally different niches (for instance, niches with the same cell type composition but different arrangements are treated as negative pairs) and to recognize identical niches in varying external environments (positive pairs). Our model employs subgraph pooling to generate niche embeddings, which can obscure the topological structure within the niche or subgraph. The introduction of contrastive learning effectively mitigates this issue.

Second, the absence of pooling also severely decreases the model's performance. We argue that node embedding methods are ill-suited for the niche query problem. This is primarily due to the fact that, while the information aggregated in node embeddings considers the microenvironments surrounding each cell, the signal diminishes with increasing distance from neighboring cells to the central cell. Given that a spatial niche comprises dozens of cells, it is challenging to identify a

Table 4: Results for Subgraph Wasserstein Distance on Mouse Olfactory Bulb Tissue dataset. We report the mean value on the 2 reference samples.

| Niche Name | GraphST | SLAT | STAGATE | QueST |
|---|---|---|---|---|
| GCL_100 | 0.794 | 0.741 | 0.812 | **0.729** |
| GCL_150 | 0.800 | 0.762 | 0.883 | **0.738** |
| GCL_50 | 0.808 | 0.729 | 0.815 | **0.692** |
| GCL_MCL_EPL_100 | 0.819 | 0.835 | 0.894 | **0.803** |
| GCL_MCL_EPL_150 | 0.822 | **0.772** | 0.881 | 0.797 |
| GCL_MCL_EPL_50 | 0.839 | 0.842 | 0.926 | **0.838** |
| GL_ONL_100 | 0.918 | 0.829 | 0.865 | **0.782** |
| GL_ONL_150 | 0.848 | 0.810 | 0.870 | **0.763** |
| GL_ONL_50 | 0.906 | 0.810 | 0.887 | **0.765** |



Figure 6: Visualization of query result on example niche GL_ONL_100

Table 5: Ablation study for batch discriminator, contrastive learning and pooling on QueST model, where No-X means the QueST model without the X module. We report mean Pearson Correlation on the 11 reference samples of the DLPFC dataset.

| Niche Name | No-Batch | No-Contrastive | No-Pooling | QueST |
|---|---|---|---|---|
| Layer3_Layer4_Layer5_100 | 0.302 | 0.316 | 0.260 | **0.383** |
| Layer3_Layer4_Layer5_200 | 0.402 | 0.263 | 0.306 | **0.620** |
| Layer3_Layer4_Layer5_50 | 0.524 | 0.272 | 0.369 | **0.585** |
| Layer4_100 | **0.540** | 0.258 | 0.226 | 0.535 |
| Layer4_200 | 0.527 | 0.259 | 0.343 | **0.535** |
| Layer4_50 | 0.508 | 0.248 | 0.213 | **0.529** |
| Layer5_Layer6_100 | 0.647 | 0.294 | 0.496 | **0.802** |
| Layer5_Layer6_200 | 0.498 | 0.393 | 0.280 | **0.743** |
| Layer5_Layer6_50 | 0.549 | 0.399 | 0.389 | **0.714** |

dominant cell. Consequently, using the central cell's node embedding as the representative for the entire niche can introduce significant bias when calculating the similarity between different niches.

Lastly, the removal of the batch discriminator has a relatively minor impact on the model's performance. This may be attributed to the relatively small batch effects present in the DLPFC dataset. Nonetheless, the overall performance indicates that the batch discriminator contributes to more accurate niche similarity measurements across different samples.

## 5 DISCUSSION AND CONCLUSION

In this study, we introduce the **Niche Query Problem** and develop QueST, a novel and specialized framework for querying spatial niches across diverse spatial transcriptomics samples. QueST employs a subgraph contrastive learning approach to explicitly learn niche representation, incorporates an adversarial training framework to perform batch correction, and finally uses cosine similarity between latent representation vectors to perform spatial niche query. Experimental results demonstrate the efficacy of these critical modules and validate QueST's capability for accurately querying spatial niches in various settings. To the best of our knowledge, QueST is the first graph deep learning model tailored for querying spatial niches in spatial transcriptomics data. We believe that with the increased volume of spatial transcriptomic data, this niche query problem will be a foundation for investigating pan-cancer or cross-species studies, and both the QueST method and the query benchmark we developed will lay the groundwork for addressing this problem.

Certain limitations exist for QueST and further work is desired to improve the framework. At present, QueST requires computing $k$-hop subgraphs with a fixed $k$ to extract spatial niches on reference samples for comparison with the query niche. Handling niches with various sizes and shapes simultaneously on reference samples is a potential future direction for improvement. Furthermore, the graph auto-encoder architecture limits the scale of the model. Introducing novel architecture such as transformers may help scale up the model for larger pre-training and querying tasks on datasets with millions of cells or even more.

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

# A    APPENDIX

## A.1    DETAILED DESCRIPTION OF DATASETS

The DLPFC dataset contains 12 samples as summarized in Table A.1:

Table A.1: Summary of the DLPFC dataset

| Sample ID | Spot Number | Gene Number |
|-----------|-------------|-------------|
| 151507 | 4221 | 33538 |
| 151508 | 4381 | 33538 |
| 151509 | 4788 | 33538 |
| 151510 | 4595 | 33538 |
| 151669 | 3636 | 33538 |
| 151670 | 3484 | 33538 |
| 151671 | 4093 | 33538 |
| 151672 | 3888 | 33538 |
| 151673 | 3611 | 33538 |
| 151674 | 3635 | 33538 |
| 151675 | 3566 | 33538 |
| 151676 | 3431 | 33538 |

Besides, the DLPFC contains 7 cell types in total, namely Layer1, Layer2, Layer3, Layer4, Layer5, Layer6, and WM, which corresponds to different layers of cortex and the white matter. The ground truth cell type annotation is depicted in Figure A.1.

The MOBT dataset contains 3 samples as summarized in Table A.2:

Table A.2: Summary of the MOBT dataset

| Sequencing Technology | Spot Number | Gene Number |
|-----------------------|-------------|-------------|
| 10X Visium | 1185 | 5531 |
| Stereo-seq | 8762 | 5531 |
| Slide-seq V2 | 18173 | 5531 |

The MOBT dataset contains 5 cell types in total, which are EPL (External Plexiform Layer), GCL (Granular Cell Layer), GL (Glomerular Layer), MCL (Mitral Cell Layer), ONL (Olfactory Nerve Layer). We show the ground truth cell type annotation in Figure A.2.

## A.2    DETAILED GENERATION OF QUERY NICHES

In this section, we describe how the query niches are selected in detail.

For the DLPFC dataset, we choose sample 151507 as the query sample, upon which we define the query niches. We propose the following three cell-type patterns:

1. Layer4;
2. Layer5 and Layer6;
3. Layer3, Layer4 and Layer6.

With the niche size determined as 50, 100, and 200 cells beforehand, we select the region that satisfies these 9 different requirements respectively. For each setting, we first randomly determine an anchor cell for the query niche and expand this subgraph with its k-hop neighbors until the subgraph satisfies the size requirements. The query niches selected are visualized in Figure A.3.

For the MOBT dataset, we choose the Stereo-seq sample as the query sample and define query niches on it. We choose the following cell-type patterns:

1. GCL;

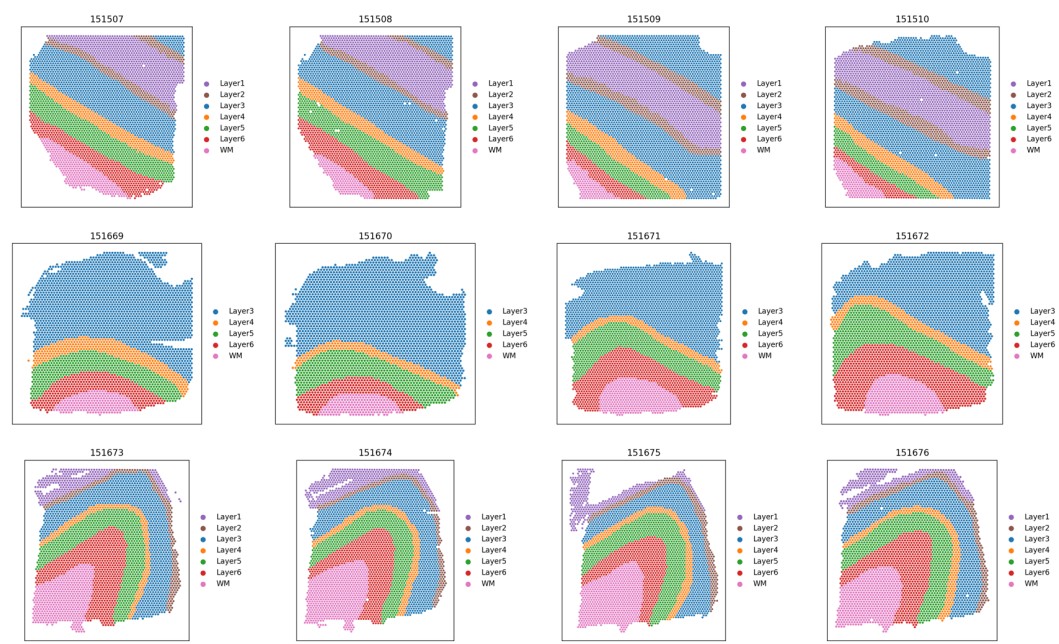

Figure A.1: ground truth cell type annotation on DLPFC dataset

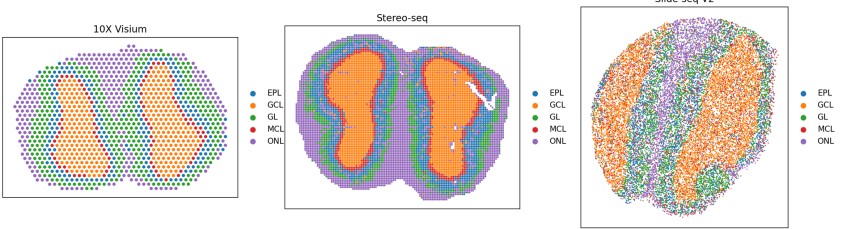

Figure A.2: ground truth cell type annotation on MOBT dataset

2. GL and ONL;

3. GCL, MCL, and EPL.

We then set out to select appropriate regions with the same niche size requirements. Since the cell type distribution in this tissue is not as smooth as in the DLPFC dataset, it is hard to find a region that satisfies the cell type requirements. Instead, we choose regions that contain the required cell types with a proportion over 95%. The query niches of the MOBT dataset are shown in Figure A.4.

### A.3 IMPLEMENTATION DETAILS FOR QUEST MODEL

Here we present the default configuration for QueST in Table A.3:

where the fix ratio refers to the fixed number of niches when generating positive and negative pairs, and the negative pair shuffle ratio is used to determine whether a shuffled subgraph can be sampled as a negative pair. QueST is implemented via PyTorch 2.2.1 and PyTorch Geometric 2.5.2. All experiments are done on an Nvidia A100-80G GPU.

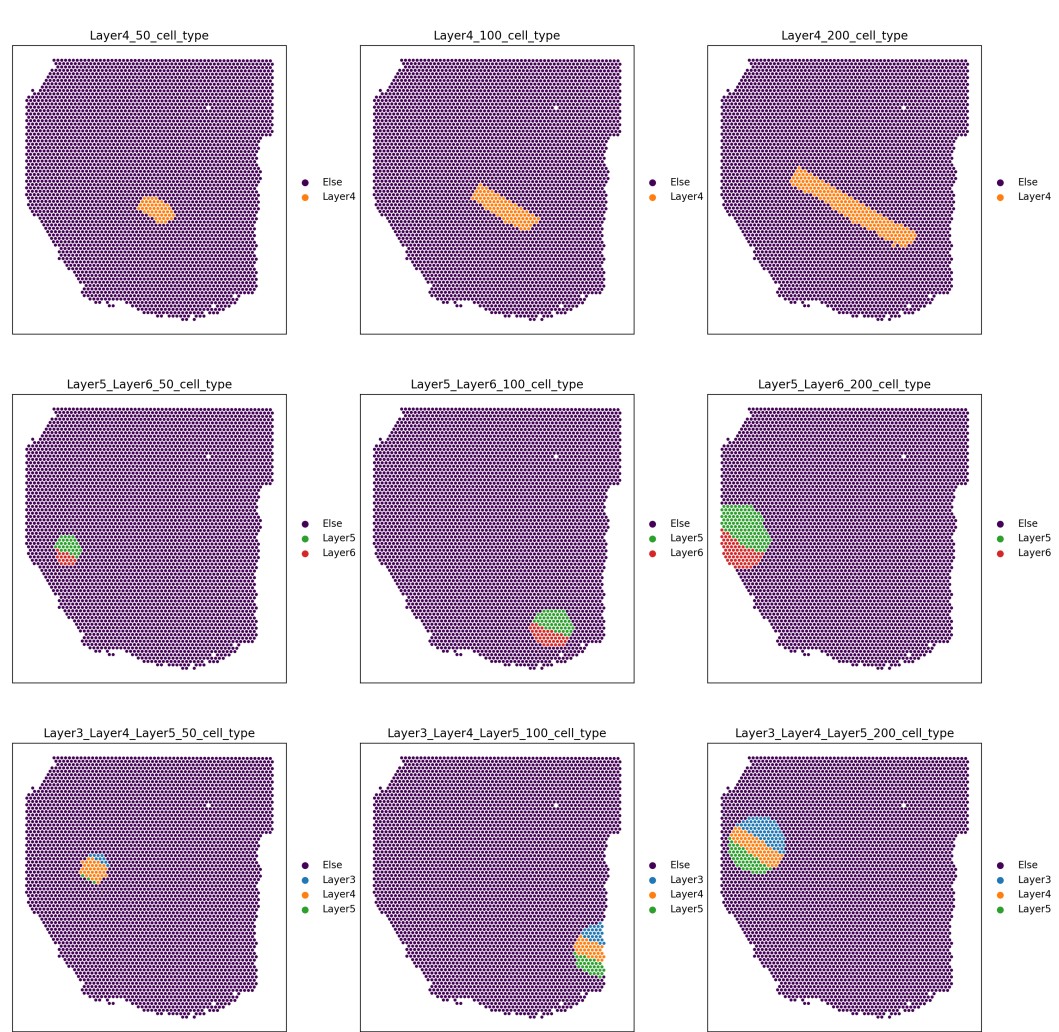

Figure A.3: Query niches of the DLPFC dataset

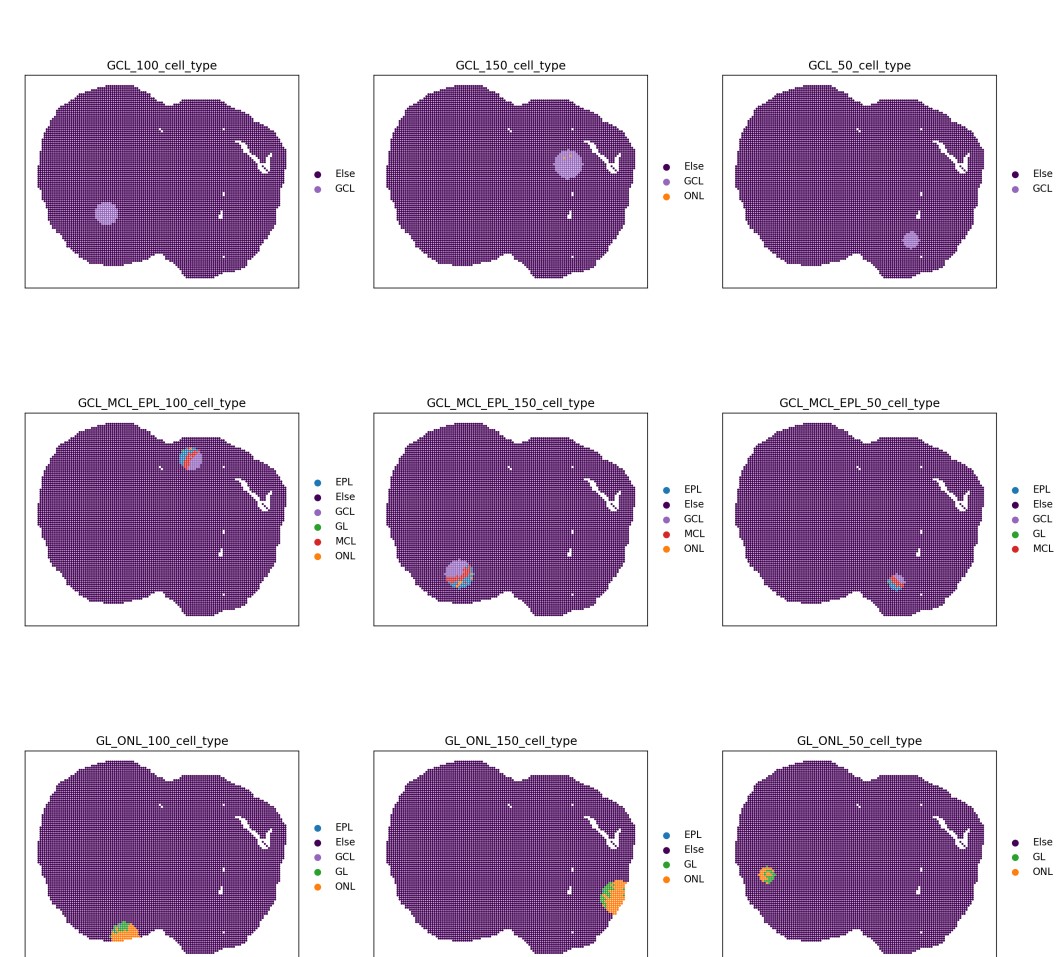

Figure A.4: Query niches of the MOBT dataset

Table A.3: Default configuration for QueST

| config | value |
|---|---|
| gnn type | GIN |
| $\epsilon$ of GIN | 0 |
| encoder gnn layer number | 3 |
| decoder gnn layer number | 1 |
| bottleneck dimension | 32 |
| k | 3 |
| fix ratio | 2% |
| negative pair shuffle ratio | 25%-75% |
| optimizer | Adam |
| learning rate | 0.001 |
| weight decay | 5e-4 |

## A.4 EXTENDED FIGURES

In this section, we present the results of the niche Layer5_Layer6_100 on all samples of the DLPFC dataset, and GL_ONL_100 on all samples of the MOBT dataset. For results of the other niches, please refer to the supplementary materials.

### A.4.1 PEARSON CORRELATION ON THE DLPFC DATASET

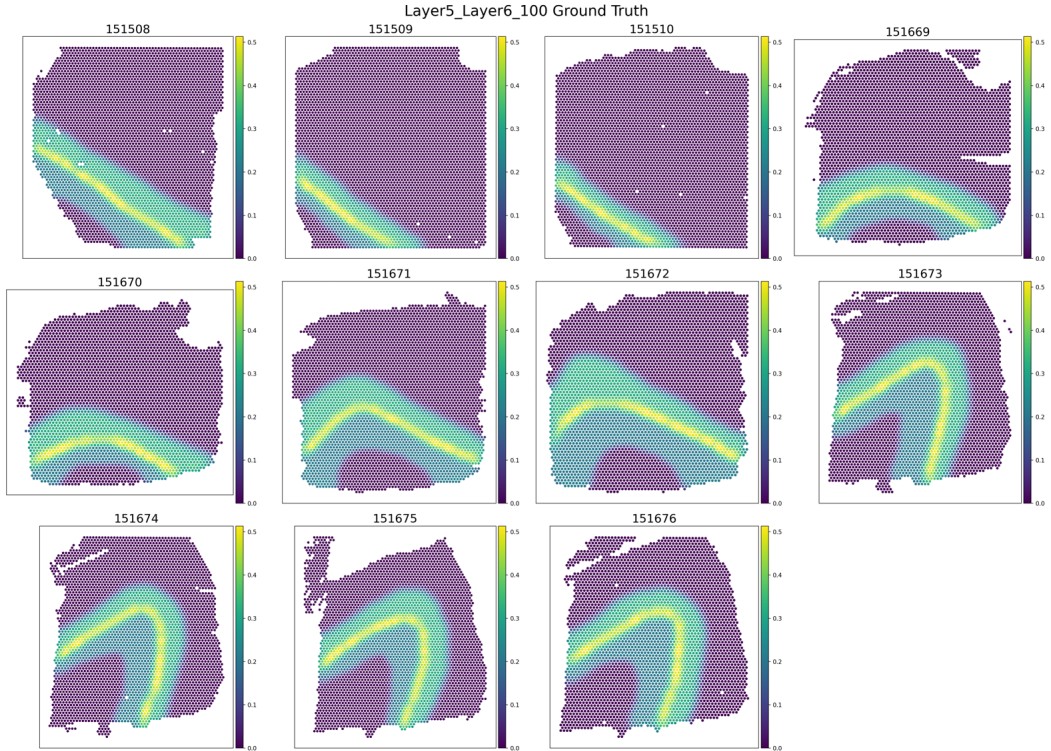

Figure A.5: Pearson Correlation of Ground Truth on the DLPFC dataset

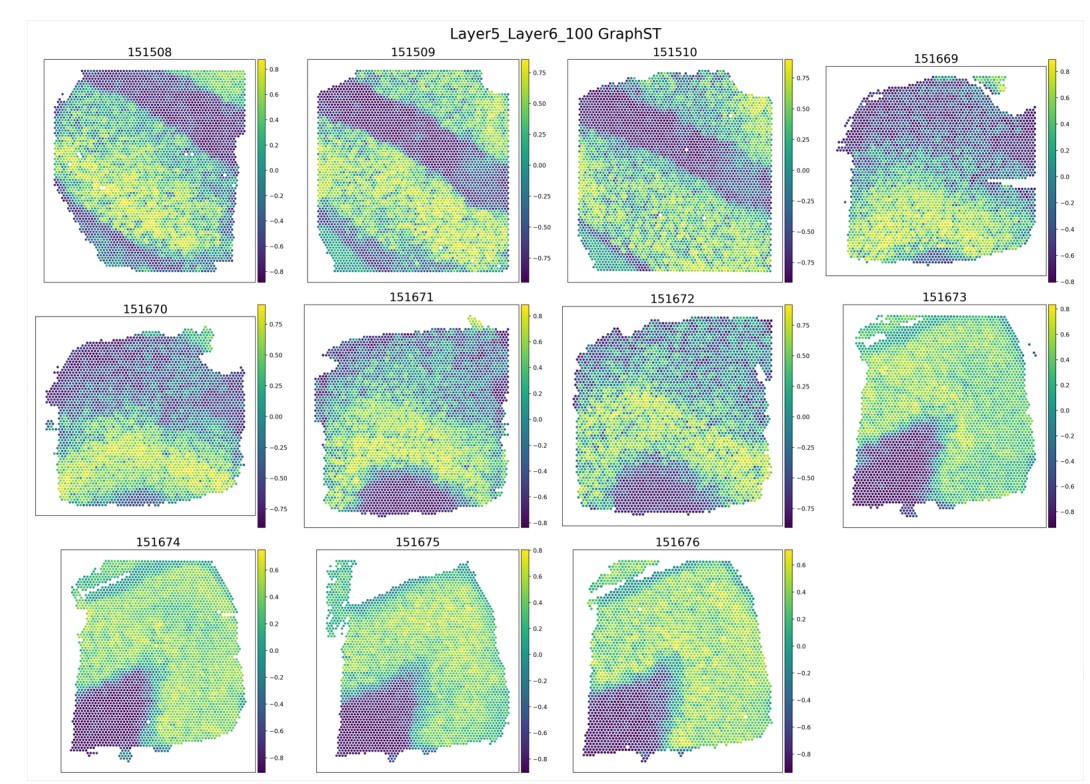

Figure A.6: Pearson Correlation of GraphST on the DLPFC dataset

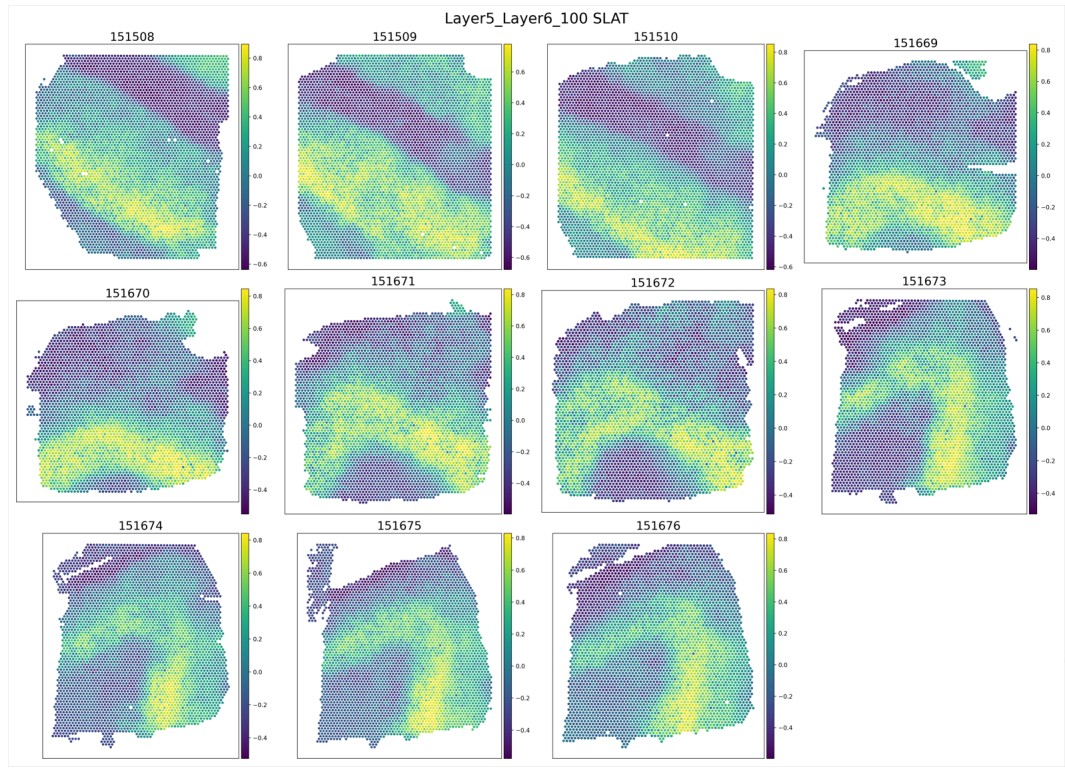

Figure A.7: Pearson Correlation of SLAT on the DLPFC dataset

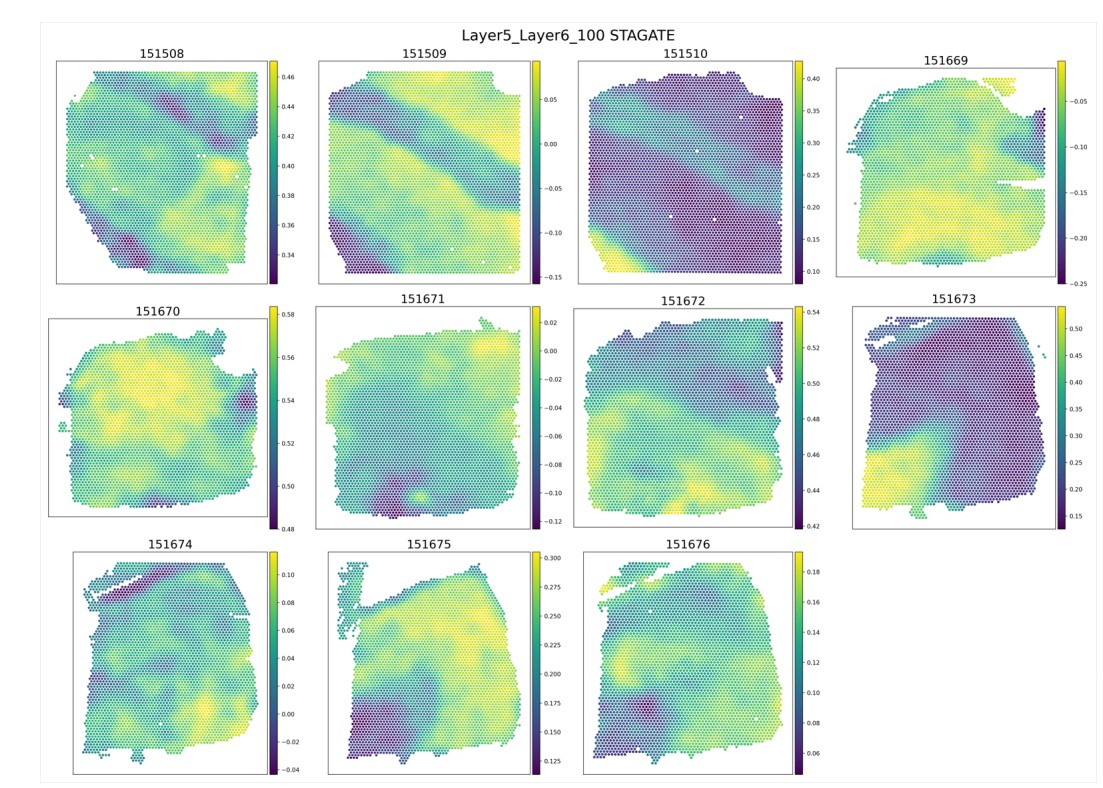

Figure A.8: Pearson Correlation of STAGATE on the DLPFC dataset

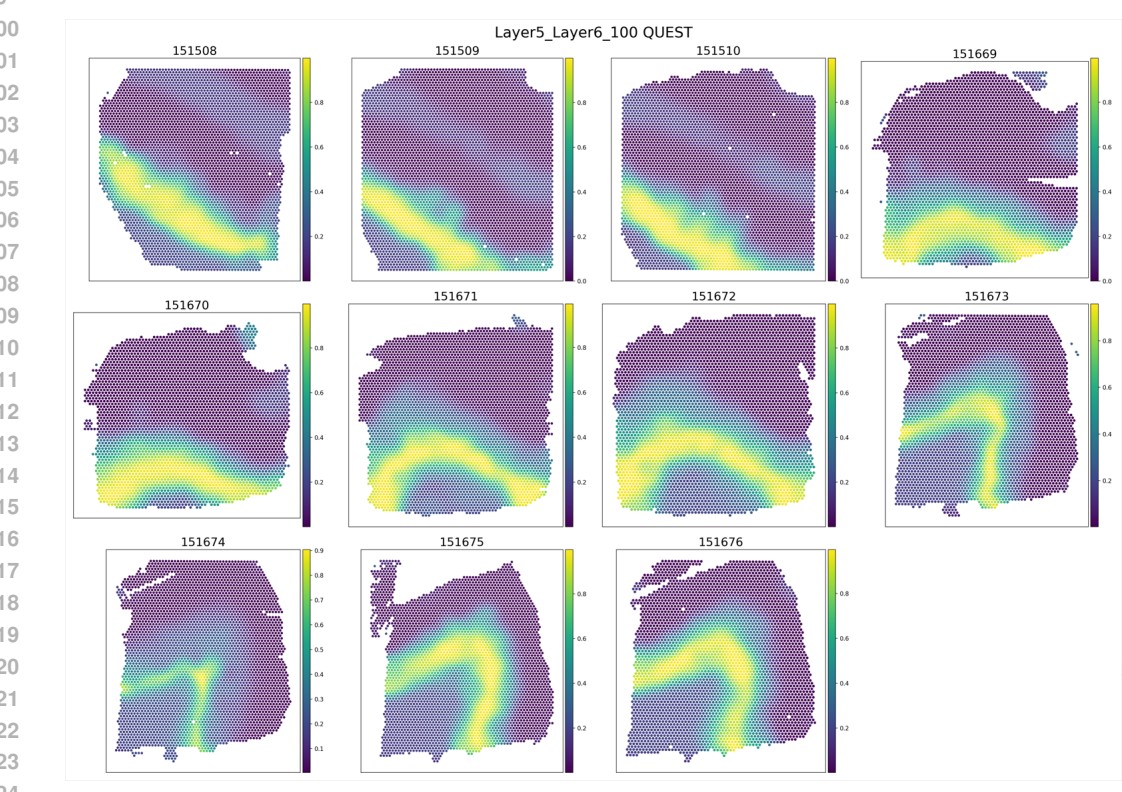

Figure A.9: Pearson Correlation of QUEST on the DLPFC dataset

### A.4.2 PEARSON CORRELATION ON THE MOBT DATASET

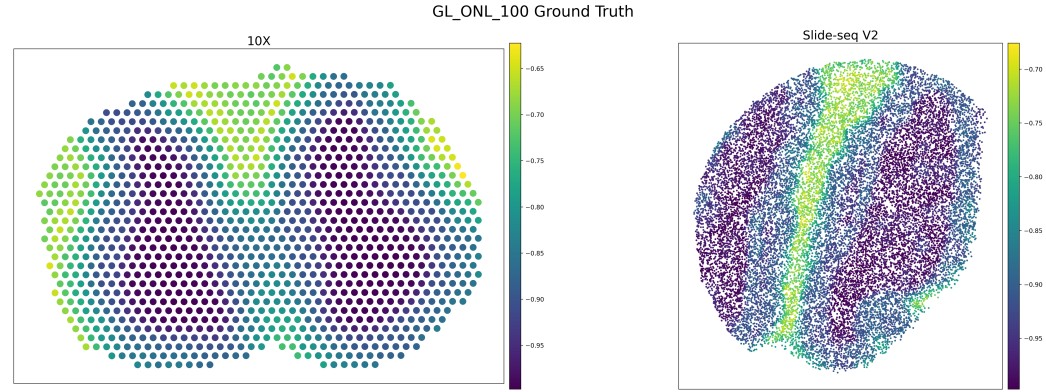

Figure A.10: Pearson Correlation of Ground Truth on the MOBT dataset

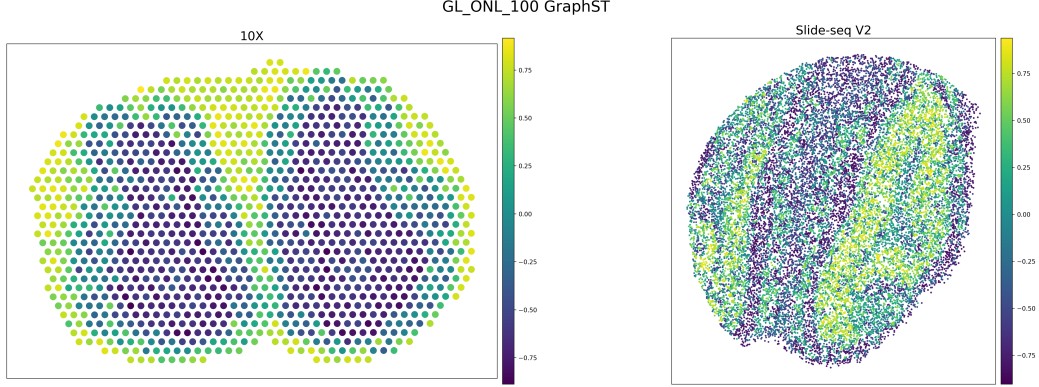

Figure A.11: Pearson Correlation of GraphST on the MOBT dataset

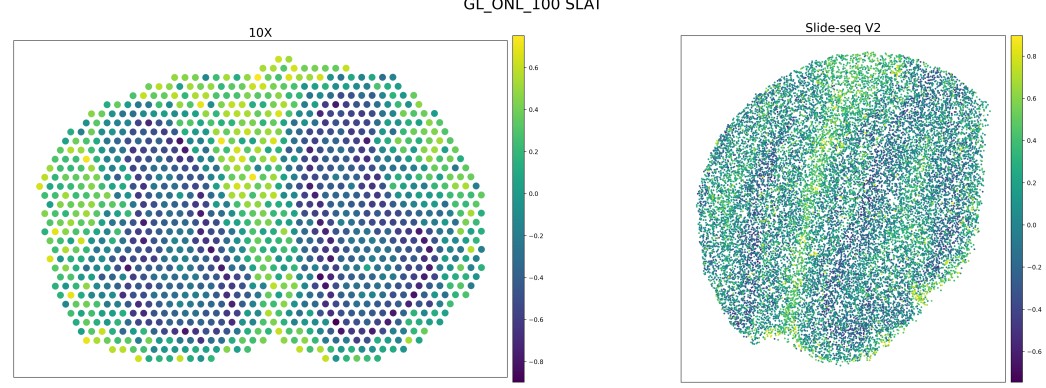

Figure A.12: Pearson Correlation of SLAT on the MOBT dataset

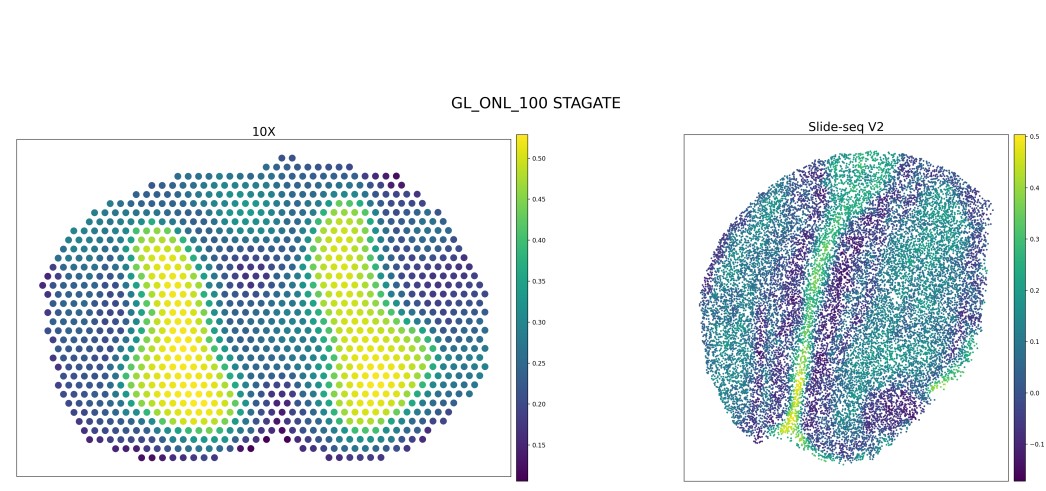

Figure A.13: Pearson Correlation of STAGATE on the MOBT dataset

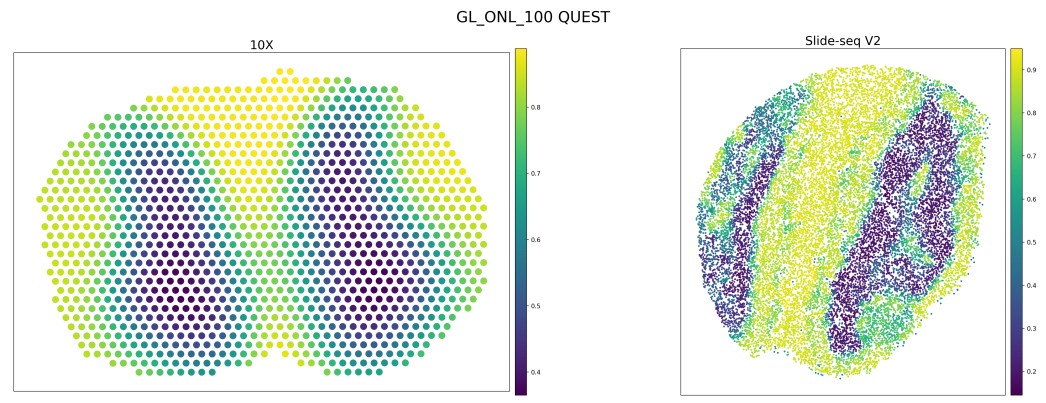

Figure A.14: Pearson Correlation of QUEST on the MOBT dataset

