# OpenReview forum: "QueST: Querying Functional and Structural Niches on Spatial Transcriptomics data via Contrastive Subgraph Embedding"
_ICLR.cc/2025/Conference — ICLR 2025 Conference Withdrawn Submission_

### Official Review · Reviewer_UpiL · 2024-10-29

**Soundness:** 2
**Presentation:** 3
**Contribution:** 2
**Rating:** 3
**Confidence:** 5

**Summary:**

The authors assert that, while identifying similar niches across samples is a critical challenge, current methods do not address the learning of niche representations or the alignment of similar niches.
To address this challenge, the authors propose a niche representation learning strategy that computes niche embeddings by pooling neighboring node embeddings in a graph, optimized through a contrastive learning approach to distinguish whether two niche representations form positive pairs.
Additionally, the authors propose an adversarial training approach that predicts the batch origin of each niche embedding, effectively removing batch effects to enable niche retrieval across different samples.
Through experiments on DLPFC and MOBT datasets, they demonstrate the effectiveness of their proposed model.

**Strengths:**

- This is the first work to focus on learning niche representations and retrieving related niches.
- The authors firstly create a niche querying benchmark based on cell-type annotation.
- The proposed model demonstrates state-of-the-art performance with a simple architecture.

**Weaknesses:**

**Major**
1) This paper lacks sufficient experimental justification for the proposed architecture, as there is no analysis provided for the model.
- To validate the necessity of niche representation, the authors should conduct ablation studies comparing the use of niche representations with using only spot representations (before pooling)
- To validate the effectiveness of the proposed losses, the authors need to conduct ablation studies or sensitivity analyses for each loss function.
- To justify the claim that adversarial training can help alleviate batch effects, the authors need to evaluate their niche representations using metrics related to batch removal, such as iLISI and kBET. Additionally, they should visualize their t-SNE or UMAP outputs by coloring based on batch effects to assess whether the representations are mingled regardless of batch.

2) The benchmark construction process is not clear.
- Among the 12 samples in the DLPFC dataset, the authors select 151507 as the query sample; however, the rationale for this selection is not provided. If there is a reason for this choice, the authors should elaborate on it. If not, I recommend using each sample as a query and averaging the results for reporting.
- Similarly, the authors choose layers 3, 4, and 5 among the 7 annotated cell types without providing justification. They should clarify the reasoning behind this selection or expand the set for a fair comparison.
- The MOBT datasets also need justification regarding both the queried sample and niche selection.

**Minor**
- It would enhance understanding of the model's architecture if the authors represented the defined notations from the scripts in the figure.

**Questions:**

In the third paragraph of the introduction, the authors discuss the necessity of analyzing spatial niches across multiple samples. It would be beneficial to include a simple analysis result using the queried outputs from the proposed model to demonstrate the biological significance of these results in relation to this explanation.

---

### Official Review · Reviewer_A3q8 · 2024-11-01

**Soundness:** 2
**Presentation:** 2
**Contribution:** 2
**Rating:** 5
**Confidence:** 3

**Summary:**

In this paper, the authors aim to query spatial niches within multicellular tissue samples, specifically for spatial transcriptomics data. The authors mentioned that existing methods focus on cell-level embeddings, limiting the understanding of spatial niches, especially across diverse samples. Therefore, QueST, a novel model for querying spatial niches in spatial transcriptomics data, is proposed to accurately query spatial niches across different datasets. Specifically, QueST introduces a subgraph contrastive learning strategy that captures niche-level representations. And, the authors utilize ‘Adversarial Batch Effect Removal’, which incorporates an adversarial training framework to eliminate batch effects, ensuring consistent cross-sample niche comparisons. In the experiment, QueST was tested on human and mouse datasets and demonstrated superior performance over existing graph representation methods in identifying similar niches across samples.

**Strengths:**

1.	It is very interesting to combine ideas from graph contrastive learning with spatial data. QueST introduces a subgraph contrastive learning strategy that captures niche-level representations, and incorporates an adversarial training framework to eliminate batch effects, ensuring consistent cross-sample niche comparisons.
2.	The paper is well-supported by rigorous experiments on benchmark datasets (human DLPFC and mouse MOBT). The comparison against state-of-the-art models like GraphST, SLAT, and STAGATE highlights QueST’s improved performance, especially in handling batch effects and accurately querying niches across different samples.
3.	The paper is clearly written, with each component of the QueST model explained in a step-by-step manner. And, the authors used plenty of figures and tables to clarify the model’s effectiveness.

**Weaknesses:**

1.	The model structure appears relatively simple, with limited technical sophistication. Overall, it lacks novelty in its architecture design. For example, in terms of adversarial learning, there doesn't seem to be a substantial contribution to the model's technical components, as it primarily relies on the existing adversarial auto-encoder framework without introducing significant modifications or advancements.
2.	Parameter choices are not clearly explained. For example, the weights assigned to different loss components (1, 0.1, 0.1) lack a rationale, and no experiments are provided to justify these specific values. An ablation study to demonstrate the effects of varying these weights on model performance is necessary.
3.	There are some typos and formatting issues in the paper. For instance, on page 6, in the sentence “For the WWL Graph Kernel, We set...," the word “We” should be lowercase. Additionally, punctuation is missing after each formula and before the “where” in explanations, which affects the overall readability of the paper.
4.	Clarity: The key challenges mentioned in the Abstract and Introduction are not clearly articulated. These challenges do not highlight the need for QueST. For example, the rationale for considering on the "niche level" is not clearly explained. Although considerable text is dedicated to discussing the challenges associated with "considering the niche level," there is a lack of thorough and compelling justification for why this consideration is necessary. The importance of a niche-level approach is not effectively established in the text.
5.	Figure 2 does not clearly illustrate the model structure. The diagram lacks detailed annotations, and the symbols are not explained, which hinders the model's interpretability. Specifically, the diagram uses overly generic labels, such as “pooling,” for the innovation components without providing enough detail on the specific role each component plays in this paper. Additionally, the data flows between components in the model are not annotated to clarify their meaning, making it difficult to understand the function of each component and the workflow of the entire model.

**Questions:**

See weakness.

---

### Official Review · Reviewer_fLet · 2024-11-04

**Soundness:** 2
**Presentation:** 2
**Contribution:** 1
**Rating:** 3
**Confidence:** 5

**Summary:**

This paper addresses the limitations of previous studies that overlooked the influence of the niche (i.e., surrounding environment) on the behavior of cells, even within the same cell type. The study is motivated by the challenges of integrating samples from diverse environments, particularly in managing batch effects. To capture representations of these niches, the authors propose a method called QueST, which leverages Deep Graph Infomax (DGI) and adversarial training.

The approach modifies the DGI structure by adding node (spot) representations with a pooling layer to extract niche embeddings, which are then fed into a discriminator. This modification creates a contrastive learning framework to learn niche embeddings effectively.
Subsequently, the model is trained to reduce batch effects using adversarial training that incorporates batch information, ensuring that the learned representations are robust across different batches.

QueST was evaluated on established benchmarks using human (DLPFC) and mouse (MOBT) datasets, demonstrating its superiority over state-of-the-art graph representation learning methods in performing accurate niche queries.

**Strengths:**

* To quantitatively evaluate the Niche Query Problem, various metrics were defined using graph matching and similarity measures.
* It is the first graph-based deep learning model specifically designed for querying spatial niches in spatial transcriptomics data

**Weaknesses:**

* The motivation is unclear regarding how the proposed niche representation differs from cross-sample similarity and how it can be utilized differently from cell-cell alignment. Including a biological definition of "niche" and an explanation of niche alignment applications in the biology domain in the introduction would be beneficial.

* While batch effect is emphasized as a key issue in the motivation, there is a lack of experiments demonstrating improvement in this area. It would be advantageous to assess batch correction effectiveness using metrics such as iLISI, kBET, Graph Connectivity, and visualize corrected batch representation space with UMAP or t-SNE. (refer to Q2)

* The use of sampling for query samples and query niches poses a risk of selective experimental results, potentially compromising the validity of the findings. It would be necessary to validate the rationale behind selecting specific niche layers such as 'layer4,' 'layer5_layer6,' and 'layer3_layer4_layer5,' or, if there are biologically significant core niches, to assess them explicitly. Additionally, evaluating entire slices instead of relying on sampling would further strengthen the analysis.

* The paper is not self-contained, as it omits definitions for terms and equations related to 'Best Niche Match Accuracy,' which are essential for a complete understanding of the results. Although equations (2) and (3) are derived from the 'Wasserstein Weisfeiler-Lehman (WWL) Graph Kernel method' [1],  it is necessary to define all relevant notations, including $\mathbf{a}, v, h, \mathcal{N}$, hash, and WassersteinDistance.

* A more comprehensive comparison with diverse baselines is needed. At a minimum, models for Spatial Transcriptomics based on DGI [2] should be included in the comparisons.

[1] Wasserstein weisfeiler-lehman graph kernels. Advances in neural information processing systems, (2019).

[2] Identifying multicellular spatiotemporal organization of cells with SpaceFlow, Nature Communications (2022).

**Questions:**

* Although the reconstruction loss may have a minimal effect, it raises the question of why reconstruction is performed with the cell rather than the niche, as this approach appears misaligned with the paper’s primary motivation. Could you clarify the rationale behind this choice?

* Could experiments be added to evaluate batch correction effectiveness, including metrics like batch embedding space, iLISI, kBET, and Graph Connectivity?

* As I understand it, PASTE and STAligner are capable of handling multiple samples per training run. The following statement seems applicable only to scSLAT:

    > Furthermore, these methods are limited to aligning only two samples per training run, making them impractical for analyzing large cohorts of spatial transcriptomics samples.

    Please clarify further if my understanding is incorrect. Otherwise, this sentence should be revised to specifically refer to scSLAT, or exclude it from a general limitation for PASTE, STAligner, and scSLAT.

---

### Note · Authors · 2024-11-15

**Comment:**

We thank the reviewers for their valuable and constructive feedback. After careful consideration, we have decided to withdraw the submission from ICLR.

**Withdrawal Confirmation:**

I have read and agree with the venue's withdrawal policy on behalf of myself and my co-authors.